# Temporal gating of nuclear import: How Merkel cell polyomavirus exploits the cell cycle for nuclear entry

Karen Wang[1], Adrienne N. Eady[1], Isabel Amaya[1], Alina Stanczak[1], Chelsey C. Spriggs [1,2,3]*

**1** Department of Cell and Developmental Biology, University of Michigan Medical School, Ann Arbor, Michigan, United States of America, **2** Department of Microbiology and Immunology, University of Michigan Medical School, Ann Arbor, Michigan, United States of America, **3** Life Sciences Institute, University of Michigan, Ann Arbor, Michigan, United States of America

* cspriggs@med.umich.edu

## Abstract

Merkel cell polyomavirus (MCPyV) is a small, DNA tumor virus that is causally linked to an aggressive form of human skin cancer called Merkel cell carcinoma. MCPyV is the only polyomavirus definitively shown to cause cancer in humans, yet little is known about how it establishes infection in target cells. In this study, we report an unconventional mechanism by which MCPyV enters the host cell nucleus, where viral genome replication occurs. We demonstrate that, unlike other known polyomaviruses, MCPyV does not require the nuclear pore complex during entry. Instead, it takes advantage of cell cycle-dependent nuclear envelope breakdown to deliver its genetic material into the nucleus. We further show that the VP1 major capsid protein is sufficient to facilitate this process. Overall, our findings reveal a novel mechanism of polyomavirus nuclear entry and provide insight into the diverse mechanisms that these viruses use to cause infection.

## Author summary

Merkel cell polyomavirus (MCPyV) causes a rare but deadly form of skin cancer called Merkel cell carcinoma. However, many aspects of its infectious life cycle are unclear, including how it reaches the nucleus for viral replication. Our study demonstrates that unlike many viruses that utilize the nuclear pore complex during entry, MCPyV bypasses this conventional nuclear import pathway and instead takes advantage of nuclear envelope breakdown during mitosis to enter the nucleus. This unique adaptation highlights a key difference between MCPyV and other polyomaviruses that will improve our understanding of its pathogenesis and aid in the development of targeted therapeutic strategies against MCPyV-associated malignancies.

**Data availability statement:** The authors confirm that all data underlying the findings are fully available without restriction. All relevant data are within the paper and its Supporting information files.

**Funding:** National Institute of General Medical Sciences, R00GM141365 to CCS, Burroughs Wellcome Fund, PDEP 1021142 to CCS. The funders had no role in study design, data collection and analysis, decision to publish, or preparation of the manuscript.

**Competing interests:** The authors have declared that no competing interests exist.

## Introduction

Polyomaviruses (PyVs) are small, non-enveloped DNA viruses that cause debilitating diseases in humans including hemorrhagic cystitis and nephropathy by BKPyV, progressive multifocal leukoencephalopathy by JCPyV, and the often-fatal Merkel cell carcinoma (MCC) by Merkel cell polyomavirus (MCPyV) [1]. The incidence of MCC has tripled over the last two decades [2], but the development of more effective therapies has been limited, in part, by an incomplete understanding of MCPyV biology, including its precise host cellular tropism, route of transmission, and infectious life cycle.

MCPyV infection is widespread amongst the human population and its viral DNA has been detected in several tissue types but most frequently in the skin [3]. It is well established that clonal integration of the MCPyV genome into host cellular DNA precedes MCC development in most cases [4]. However, Merkel cells are unlikely to be the primary target of MCPyV infection. Although dermal fibroblasts have been shown to support productive viral infection in culture [5], how initial infection by MCPyV leads to MCC and whether dermal fibroblasts are the primary reservoir of viral replication are still debated. Therefore, identifying the viral and host cellular requirements that are fundamental for MCPyV infection is essential for understanding the role of MCPyV in MCC pathogenesis.

Initial studies on MCPyV entry have determined that it first binds to sulfated glycosaminoglycans and sialic acid motifs on the host cell surface [6]. The virus then enters the cell using caveolar/lipid raft-dependent endocytosis [7] and traverses the complex endomembrane system to deliver its genetic material into the nucleus where DNA transcription and replication take place. As with most DNA viruses, nuclear entry is a final and decisive step in the MCPyV entry pathway, yet the exact mechanisms by which it enters the host cell nucleus are poorly characterized.

Structurally, the MCPyV genome is 5.4kb in size and encased within a 45-nm icosahedral capsid shell. While it shares genetic and structural similarities with well-studied PyVs such as BKPyV, JCPyV, and simian virus 40 (SV40), several unique features distinguish it from its counterparts that may affect viral entry and cellular tropism. For instance, most PyVs encode three structural proteins: an outer VP1 major capsid protein and internal VP2 and VP3 minor capsid proteins. In contrast, MCPyV is composed of just VP1 and VP2 and lacks the genetic architecture to produce VP3 [3,8], which previous studies have shown is required for multiple steps in the PyV entry pathway including nuclear entry [9–11]. Therefore, we sought to understand how MCPyV reaches the nucleus during initial stages of infection and the contribution of the viral capsid proteins to this process.

Here, we report a novel mechanism of PyV nuclear entry through which MCPyV largely bypasses the conventional nuclear pore complex (NPC)-dependent pathway to deposit its genome into the nucleus. By combining inhibitor studies, molecular biology, and imaging techniques, we demonstrate that MCPyV instead takes advantage of the temporal dynamics of the cell cycle to enter the nucleus during mitotic nuclear envelope breakdown (NEBD). We further show that VP1 is sufficient to direct this nuclear entry pathway independent of VP2. Overall, our findings provide new insight

into the cellular requirements for productive MCPyV infection, which may ultimately prove valuable in understanding its oncogenic potential.

## Results

### COS-7 cells support robust MCPyV infection

To investigate the nuclear entry requirements of MCPyV, we utilized an established pseudovirus (PsV) system that was developed to study the entry dynamics of small DNA viruses [12,13]. MCPyV PsV particles are composed of the VP1 and VP2 capsids and harbor an internal reporter expressing green fluorescence protein (GFP) in place of the viral genome (Fig 1A,B). Since GFP is only transcribed once the PsV reaches the nucleus, it serves as a surrogate marker for successful nuclear entry. The cell type that naturally supports the MCPyV life cycle is still unclear, but recent studies have used A549 human lung epithelial cells to study MCPyV entry because they are susceptible to infection and permissive to viral replication [7,14]. We infected A549 cells with MCPyV-GFP and did observe GFP expression; however, it was slow (taking ~96 h) and required a high concentration of virus to detect a discernible signal by western blot (Fig 1C). Therefore, to optimize our experimental system we also tested the susceptibility of CV-1 and COS-7 cells (used to study SV40 entry) to MCPyV-GFP infection. While CV-1 cells displayed a similar susceptibility as A549 cells, COS-7 cells showed robust expression of GFP using 5-10x less virus (Fig 1C, D). Furthermore, GFP expression could be detected in COS-7 cells as early as 48 h post infection (hpi) by western blot and 24 hpi by qPCR (Fig 1D, E). Importantly, control experiments showed that treating cells with increasing concentrations of unencapsulated GFP reporter plasmid did not result in GFP expression, while MCPyV-GFP infection did (Fig 1F). These data further validate using the MCPyV PsV system in COS-7 cells as a reliable readout for identifying the mechanism of MCPyV nuclear entry.

### The NPC is not the primary mechanism of MCPyV nuclear entry

In cells, the canonical nucleocytoplasmic transport pathway is mediated by NPCs in the nuclear envelope that allow for the passive diffusion of ions and small molecules less than ~9 nm in diameter, and the selective passage of macro-molecules of up to 39 nm in diameter [15]. The latter requires that molecules contain a nuclear localization signal (NLS), which is recognized by importin-α/β nuclear transport receptors that chaperone the cargo through the pore [16]. In fact, many DNA viruses contain NLS sequences on their capsids that route them to the NPC [17]. MCPyV has a strong NLS predicted near the N-terminus of its VP1 outer capsid protein [8]. Therefore, as an initial strategy to identify the mechanism of MCPyV nuclear entry, we asked whether it might also use the NPC during entry.

To test this, we treated cells with importazole (IPZ), a specific inhibitor of importin-β-mediated transport, 1 h before MCPyV-GFP infection. Compared to BKPyV-GFP, a human polyomavirus that uses the NPC for entry [18], MCPyV-GFP largely retained its ability to enter the nucleus under IPZ treatment (Fig 2A). As a complementary approach, we used siRNA to knockdown (KD) importin-7 (IPO7), an importin-β subunit that was identified as an essential host factor for BKPyV entry in a whole-genome siRNA screen [19]. Compared to a scrambled control, BKPyV-GFP infection was inhibited under IPO7 KD conditions, but MCPyV-GFP infection was again unaffected (Fig 2B). To more directly assess MCPyV dependency on the NPC, we used siRNA to disrupt the overall integrity of the complex. The loss of NUP98, an FG-repeat nucleoporin, causes selective changes in NPC stoichiometry and diminishes the nuclear import of proteins with NLS sequences [20]. In concordance with our previous results, we found that NUP98 KD blocked BKPyV-GFP infection, but MCPyV-GFP was still only partially inhibited (Fig 2C). Together, these findings suggest that MCPyV may rely on a different mechanism for nuclear entry than the classical NPC-dependent pathway.

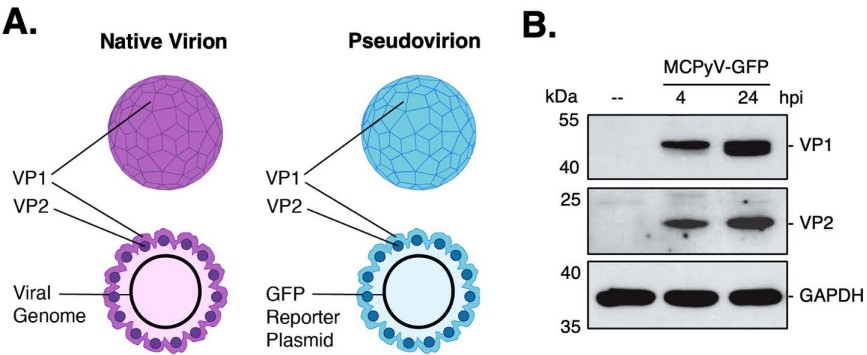

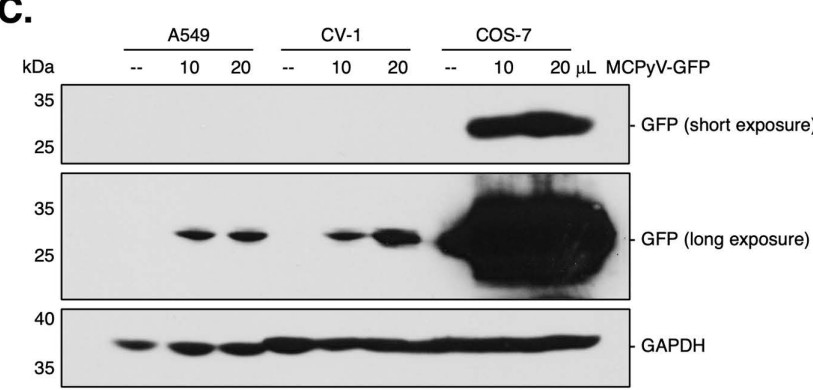

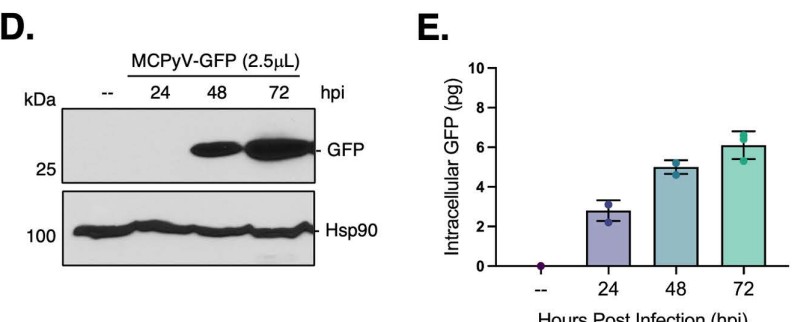

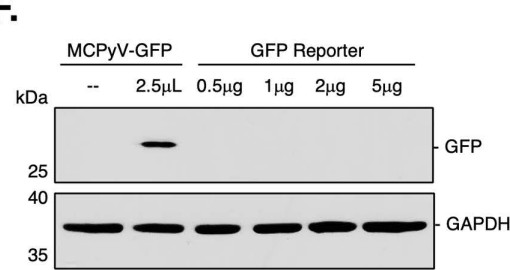

**Fig 1. Validating the MCPyV pseudovirus system in COS-7 cells.** (A) Schematic of MCPyV native virions (left) compared to pseudovirions (right). (B) COS-7 cells were infected with MCPyV-GFP for 4 or 24 h. VP1 and VP2 protein levels were assessed by immunoblotting and GAPDH was used as a loading control. (C) A549, CV-1, or COS-7 cells were infected with increasing amounts of MCPyV-GFP and harvested at 96 h post-infection (hpi). GFP

levels were assessed by immunoblotting and GAPDH was used as a loading control. (D) COS-7 cells were infected with MCPyV-GFP for 24-72 h and GFP levels measured by immunoblotting. Hsp90 was used as a loading control. (E) As in (D), except qPCR analysis of the GFP reporter plasmid DNA was performed and normalized to intracellular GAPDH. (F) COS-7 cells were either infected with MCPyV-GFP for 48 h or treated with increasing concentrations of GFP reporter plasmid as a control. Values represent means ± SD from at least three independent experiments normalized to the loading control.

## VP1 directs MCPyV nuclear entry

Our data suggests that MCPyV nuclear entry is distinct from that of BKPyV and other well-studied PyVs. This intriguing finding prompted us to determine the role of the MCPyV VP1 and VP2 capsid proteins in this process. To begin, we mutated the predicted NLS regions in VP1-ΔNLS to determine whether MCPyV PsVs can still enter the nucleus in the absence of this sequence (S1A, B Fig). We transfected either wild-type or VP1-ΔNLS into cells and confirmed that the latter is, in fact, occluded from the nucleus (S1B Fig). However, because newly replicated PyVs are assembled in the nucleoplasm, mutating the sequence prevented us generating functional MCPyV-ΔNLS to test with our PsV infection system (S1C–E Fig). In addition to the NLS sequence in VP1, most PyVs also contain conserved NLS sequences in their VP2 and VP3 capsid proteins. Interestingly, MCPyV does not encode a VP3 and there is no predicted NLS within its VP2 either [8]. As the VP2/3 NLS is required for BKPyV nuclear entry [18], we sought to investigate the significance of these structural differences on MCPyV infection.

To examine this, we generated a chimeric PsV containing MCPyV VP1 along with the BKPyV VP2/3 minor capsids to determine if the addition of a functional NLS sequence in the interior of the PsV shell would influence the MCPyV nuclear entry pathway (Fig 3A, B). After using EM to confirm the proper assembly and overall morphology of the chimeric PsV (Fig 3C), we tested the effect of IPZ on its ability to reach the nucleus. Interestingly, the chimeric PsV was still able to enter the nucleus under IPZ treatment to a similar extent as wild-type MCPyV-GFP (65.7% vs 75.6%) (Figs 2A and 3D), which suggests that the absence of an NLS sequence in the MCPyV VP2 capsid is not responsible for its distinct nuclear entry mechanism.

Accordingly, we then generated another PsV expressing only the MCPyV VP1 capsid protein to determine whether VP1 is sufficient for viral trafficking into the nucleus. Surprisingly, there was no significant difference between 'VP1 only' infection compared to wild-type or chimeric PsV infection (Fig 3E), indicating that VP1 is the dominant factor in directing MCPyV nuclear entry and not VP2/3 as seen with other PyVs.

## Cell cycle progression through mitosis is required for MCPyV nuclear entry

Since MCPyV does not appear to require conserved NLS sequences or the classical NPC-dependent nuclear import pathway for entry, we investigated alternative mechanisms by which it might gain access to the nucleus to establish infection. During cell division, cells undergo a complex and highly coordinated remodeling process that includes nuclear envelope breakdown (NEBD) at the onset of mitosis. Here, the NPCs dissociate from the membrane and the nuclear lamina depolymerizes, allowing for newly replicated chromosomes to segregate into two daughter cells [21]. Upon mitotic exit, these events are reversed and the membrane barrier between nuclear and cytoplasmic components is reformed. Interestingly, some viruses have been shown to hijack this window of opportunity to deposit their genomes into the nucleus in an NPC-independent manner [22–25].

To investigate whether MCPyV might also use mitotic NEBD to enter the nucleus, we synchronized cells with aphidicolin, a small molecule inhibitor that reversibly arrests cells at the G1/S phase border, 24 h before MCPyV-GFP infection (Fig 4A,B). Unlike IPZ treatment, we found that aphidicolin significantly blocked MCPyV-GFP infection compared to the DMSO control (Fig 4C, lane 1 vs 2). Importantly, releasing cells from this cell cycle arrest by replacing the media with DMSO 4 hpi significantly rescued this phenotype (Fig 4C, lane 3). To further probe the role of cell cycle progression in MCPyV infection, we performed a similar experiment using Ro-3306 to arrest cells at the G2/M phase border instead

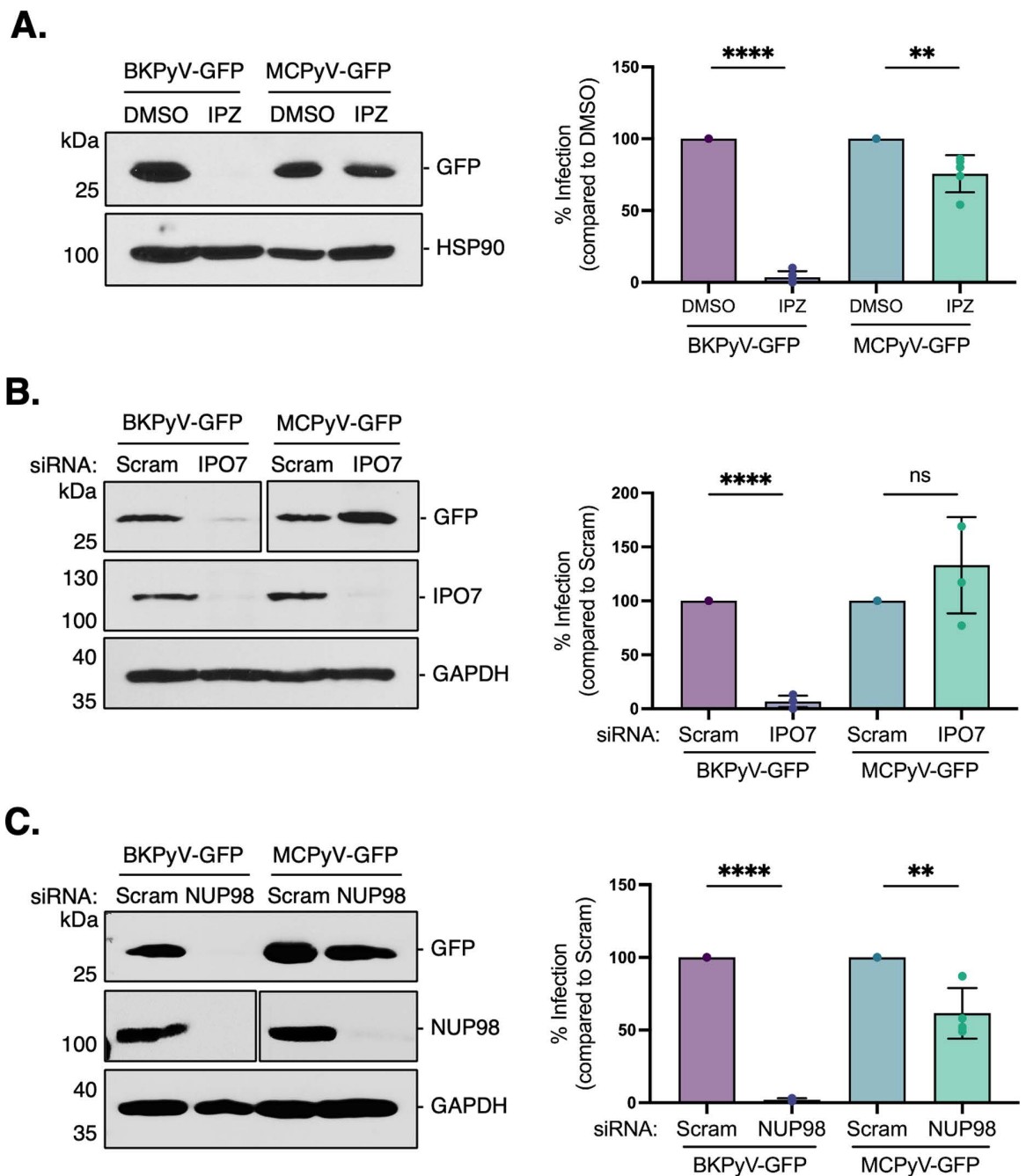

**Fig 2. MCPyV entry exhibits independence from canonical nuclear import factors.** (A) COS-7 cells were pre-treated with importazole (IPZ [10 μM]) or DMSO control for 1 h and infected with BKPyV-GFP or MCPyV-GFP for 48 h. Cells were treated with IPZ through the duration of the experiment. GFP levels were assessed by immunoblotting and Hsp90 was used as a loading control. (B) COS-7 cells were transfected with 50 nM of either scrambled control siRNA (Scram) or siRNA against importin-7 (IPO7) and infected with BKPyV-GFP or MCPyV for 48 h. GFP levels were assessed by immunoblotting and GAPDH was used as a loading control. (C) As in (B), except cells were transfected with siRNA against NUP98. Values represent means ± SD from at least three independent experiments normalized to the loading control. Statistical significance was determined using an unpaired two-tailed Student's t-test (**p ≤ 0.01, ****p ≤ 0.0001).

**A.**

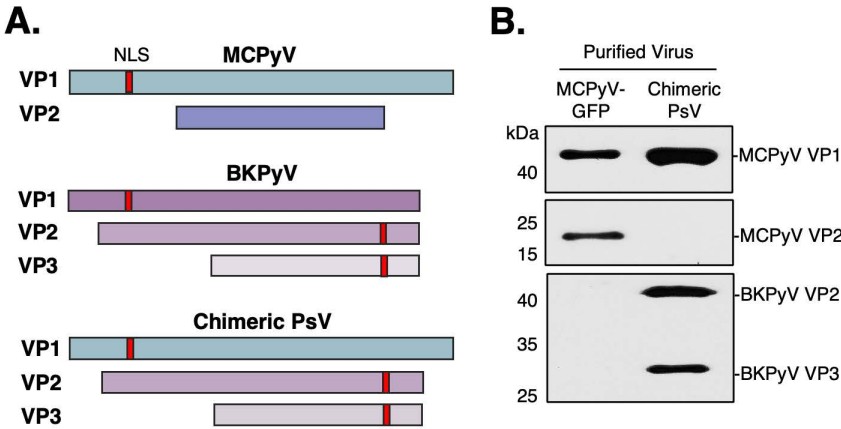

**B.**

**C.**

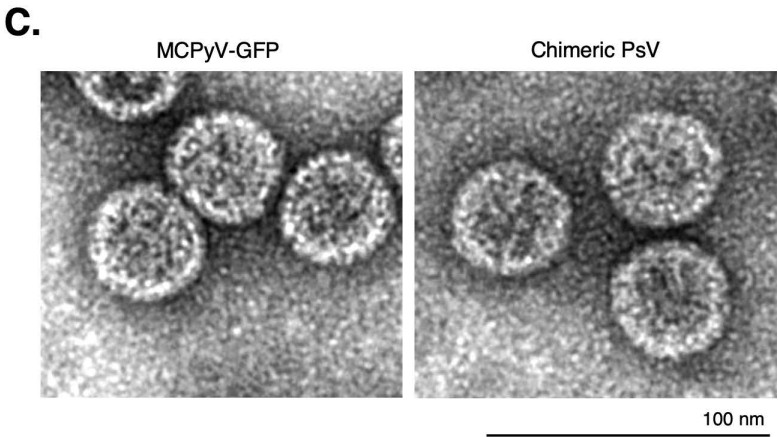

100 nm

**D.** **E.**

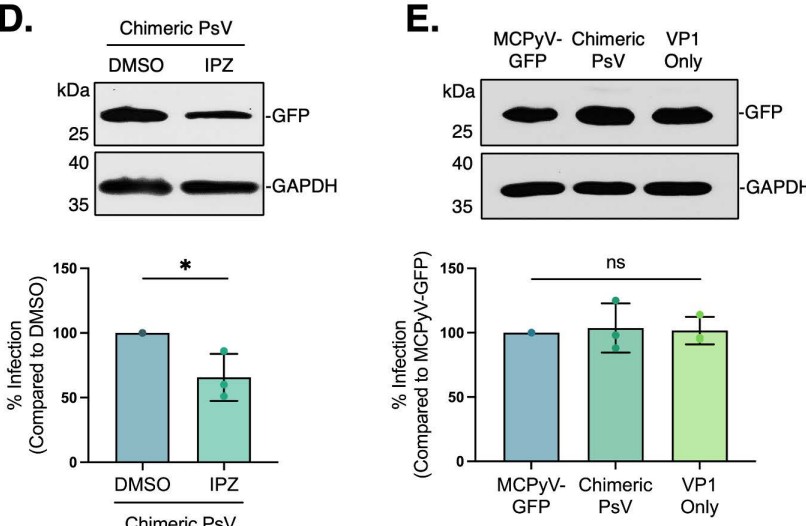

**Fig 3. VP1 directs MCPyV entry into the nucleus.** (A) Schematic showing capsid composition of MCPyV, BKPyV, and chimeric pseudoviruses (PsV). Nuclear localization signals (NLS) are depicted in red. (B) SDS-PAGE of purified MCPyV-GFP or chimeric PsV followed by immunoblotting with antibodies against MCPyV VP1 and VP2 or BKPyV VP2/3. (C) Negative-stained images of purified virions at 5000 x magnification. (D) COS-7 cells were pre-treated with importazole (IPZ [10 μM]) or DMSO control for 1 h and infected with chimeric PsV for 48h. GFP levels were assessed by immunoblotting

and GAPDH was used as a loading control. (E) COS-7 cells were infected with MCPyV-GFP, chimeric PsV or VP1 only PsV for 48 h. GFP levels were assessed by immunoblotting and GAPDH was used as a loading control. Values represent means ± SD from at least three independent experiments normalized to the loading control. Statistical significance was determined using an unpaired two-tailed Student's t-test (*p ≤ 0.05).

(Fig 4A, B). Again, we observed a block in MCPyV-GFP infection (Fig 4D, lane 1 vs 2) that was similarly reversed upon removal of the inhibitor (Fig 4D, lane 3). Together, these data suggest that cell cycle progression through mitosis may be required for the MCPyV entry pathway.

To test this hypothesis more precisely, we adapted an established experimental protocol to monitor PsV infection while strategically allowing cells to progress through single rounds of mitosis [22]. Aphidicolin blocks DNA replication in S phase, while Ro-3306 specifically inhibits CDK1 to prevent mitotic entry [26,27]. Therefore, we synchronized cells at the G1/S phase border using aphidicolin before immediately replacing it with Ro-3306 to permit the cell cycle to progress through S phase and to the G2/M phase border. Importantly, under these conditions, MCPyV-GFP infection was still blocked (Fig 4E, lane 2). This indicates that progression through S phase is not sufficient for MCPyV nuclear entry. Next, we released the cell cycle for 20 h before adding Ro-3306 to allow cells to complete a single round of mitosis. Interestingly, we now began to observe low levels of MCPyV-GFP infection (Fig 4E, lane 3). These PsV infection levels were increased when we released the cell cycle for 44 h before adding Ro-3306, thus allowing cells to progress through two complete rounds of mitosis (Fig 4E, lane 4 vs. 5). The activities of these inhibitors were confirmed to affect the cell cycle as expected under each of these conditions by flow cytometry (S2 Fig). Thus, collectively, these data suggests that cell cycle progression through mitosis is required for MCPyV infection. Importantly, Ro-3306 had no effect on BKPyV-GFP infection (Fig 4F), confirming that intracellular trafficking towards the nucleus was not globally disrupted by preventing mitotic entry. Ro-3306 did, however, block 'VP1 only' PsV infection (Fig 4G), indicating that the MCPyV VP1 capsid protein directs viral nuclear entry through this distinct cell cycle-dependent mechanism. Similar results were observed when using CV-1 cells (that do not contain SV40 large T antigen) and non-immortalized normal human dermal fibroblasts (NHDF), which may represent a more relevant cell type for natural tropism of the virus [5].

## Increasing membrane permeability is sufficient to promote MCPyV nuclear entry

Our data thus far suggest that MCPyV utilizes mitotic NEBD to enter the nucleus. We hypothesized that if MCPyV does, in fact, enter the nucleus through this process, then artificially enhancing nuclear envelope permeability in interphase cells should lead to increased infection levels. CHMP7 is part of the ESCRT-III protein complex, and is recruited to the inner nuclear membrane where it initiates the repair of large gaps in the nuclear envelope including those formed during cell division [28,29]. Therefore, to assess the role of NEBD in MCPyV nuclear entry, we used siRNA to KD CHMP7 and disrupt nuclear envelope reformation at the end of mitosis (Fig 5A,B).

To confirm that the loss of CHMP7 leads to increased nuclear envelope permeability, we transfected cells with an mCherry-NLS construct that localizes to the nucleus [30]. As expected, we observed a significant increase in cytoplasmic mCherry signal under CHMP7 KD conditions compared to the scrambled control (Fig 5C, D). We then used electron microscopy to verify that CHMP7 KD is sufficient to induce gaps in the nuclear envelope that are larger than an MCPyV virion (45-nm) (Fig 5E). Finally, we infected these cells with MCPyV-GFP and found that PsV infection was enhanced by nearly 3-fold under CHMP7 KD conditions compared to the scrambled control (Fig 5F). Taken together, these results suggest that NEBD during mitosis is a rate-limiting step in the MCPyV nuclear entry mechanism.

## Discussion

MCPyV is the only human polyomavirus definitively shown to cause cancer in humans—a rare skin cancer called Merkel cell carcinoma. However, several aspects of MCPyV biology remain unclear including how it establishes initial infection in target host cells. This study elucidates the mechanism of MCPyV nuclear entry, an essential but poorly characterized step in the entry pathways of many DNA viruses. Our findings reveal that MCPyV hijacks mitotic nuclear envelope breakdown

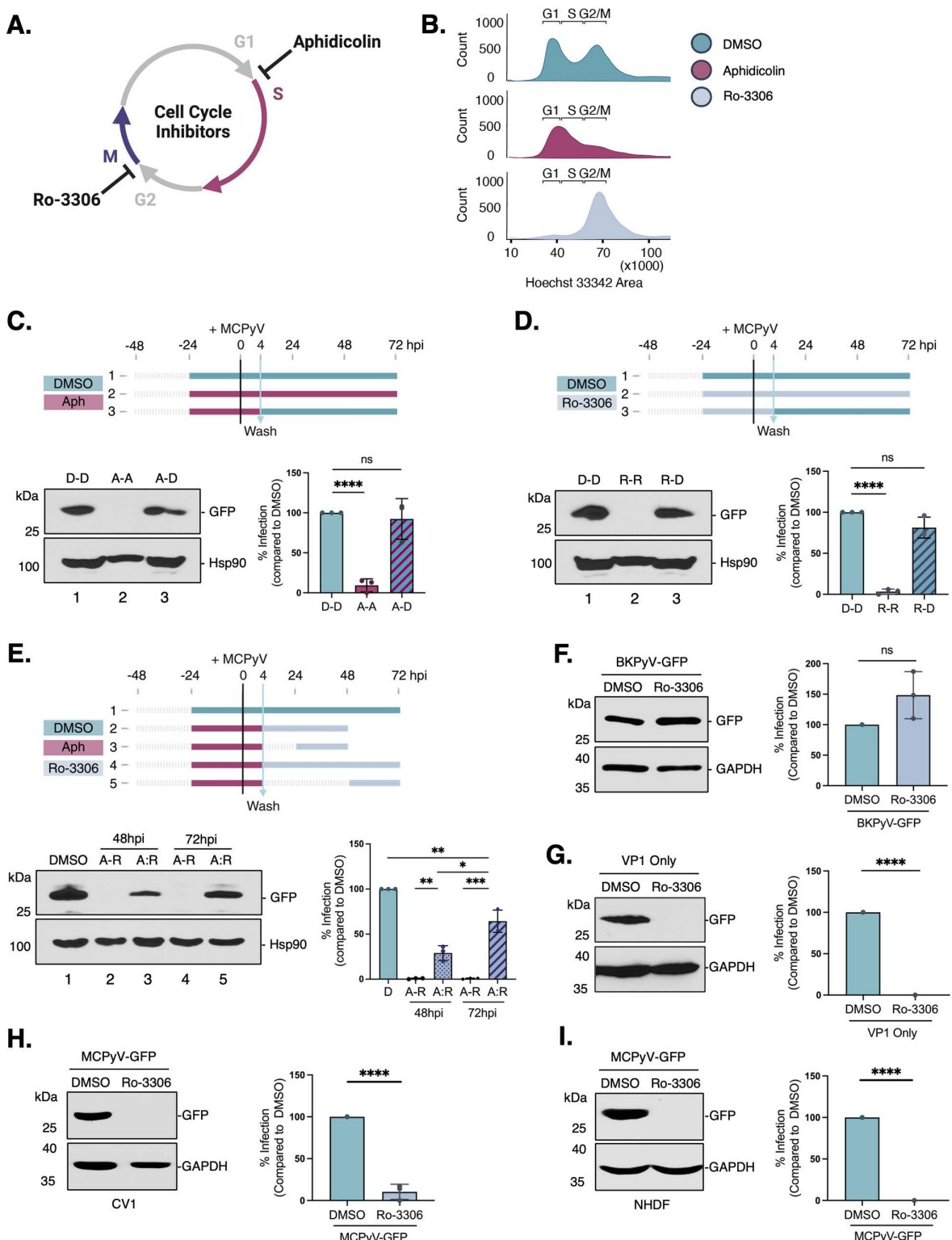

**Fig 4. MCPyV nuclear entry requires mitotic nuclear envelope breakdown.** (A) Schematic of cell cycle arrest strategy. (B) COS-7 cells were treated with either DMSO, aphidicolin (Aph [2 µM]) or Ro-3306 (9 µM) for 24 h. Cells were stained with Hoechst dye and cell cycle profiles were confirmed using flow cytometry. (C) COS-7 cells were treated with either DMSO or Aph for 24 h and infected with MCPyV-GFP. After 4 h, cells were washed and media

containing either DMSO or Aph was added again. Cells were harvested at 72 hpi and GFP levels were assessed by immunoblotting. Hsp90 was used as a loading control. (D) As in (C), except Ro-3306 was used instead of Aph. (E) As in (C), except either Ro-3306 or untreated media was added at 4 hpi for the indicated timepoints. (F) COS-7 cells were treated with Ro-3306 for 24 h and infected with BKPyV-GFP for an additional 48 h. GFP levels were assessed by immunoblotting and GAPDH was used as a loading control. (G) As in (F), except cells were infected with VP1 only PsV. (H) As in (F), except CV-1 cells were infected with MCPyV-GFP (I) As in (F), except non-immortalized normal human dermal fibroblasts were infected with MCPyV-GFP. Graphs represent averages of the means ± SD from at least three independent experiments normalized to the loading control. Statistical significance was determined using an unpaired two-tailed Student's t-test (*p ≤ 0.05, **p ≤ 0.01, ***p ≤ 0.001, ****p ≤ 0.0001).

to enter the nucleus rather than relying on the NPC. This mechanism, directed primarily by the VP1 major capsid protein, represents a novel mechanism of PyV nuclear entry and provides key insights into the infectious life cycle of this oncogenic virus (Fig 5G).

During entry, MCPyV binds to sulfated glycosaminoglycans on the host cell surface for initial attachment. The virus is then endocytosed and trafficked towards the nucleus where replication of the viral genome takes place. The mechanism by which MCPyV enters the nucleus is not known. Because other well-studied PyVs, such as SV40 and BKPyV, use the NPC for nuclear entry, we initially tested the role of this macromolecular complex in the MCPyV life cycle. While BKPyV infection was completely blocked by disrupting NPC transport, MCPyV infection remained largely intact. As we were unable to generate assembled MCPyV-ΔNLS PsVs to test in our infection system, we still cannot rule out whether this sequence in VP1 plays any role in MCPyV nuclear entry at this time. For example, it may be important for targeting the virus to the outer nuclear membrane prior to the onset of mitotic NEBD. Either way, this intriguing result suggested that MCPyV might use an NPC-independent pathway to reach the nucleus instead. Using a combination of cell cycle inhibitors, we demonstrate that MCPyV nuclear entry requires cell cycle progression through mitosis. This finding is in agreement with a previous report showing that MCPyV infection is sensitive to S phase arrest [7] and suggests that its primary cellular tropism may be an actively dividing cell.

While aphidicolin and Ro-3306 completely blocked MCPyV infection, we observed a moderate decrease in MCPyV infection under both IPZ treatment and NUP98 KD. This could indicate that while the NPC is not the primary mechanism of MCPyV nucleocytoplasmic transport, a subset of viral particles may still use the NPC for nuclear entry. However, IPZ treatment can also inhibit importin-β-mediated spindle assembly and positioning during mitosis [31,32], and NUP98 phosphorylation is critical for initiating NPC disassembly and NEBD during mitotic entry [33]. Therefore, these reductions in infection are more likely attributed to their secondary effects on cell cycle progression.

Cellular membranes and organelles undergo complex and dynamic rearrangements during cell division. To confirm that the increased nuclear envelope permeability during mitosis is required for MCPyV infection, we took advantage of an ESCRT-III-dependent repair mechanism to artificially prolong the reformation of the nuclear envelope after mitosis [28]. We hypothesized that the nuclear envelope in CHMP7 KD cells would be more permeable and, therefore, more susceptible to MCPyV infection. Interestingly, CHMP7 KD cells showed a nearly 3-fold increase in MCPyV nuclear entry compared to control cells. These results suggest that the gaps generated by CHMP7 KD, which were larger than an intact MCPyV virion, remove the physical barrier between the virus and nucleoplasm to eliminate the temporal requirement for mitosis.

Why might MCPyV have evolved to require mitotic NEBD instead of entering the nucleus through the NPC? The answer may lie in its genetic architecture and structural biology. Upon endocytosis, PyVs typically traffic to the endoplasmic reticulum (ER) where ER-resident proteins reduce and isomerize disulfide bonds between VP1 pentamers on the capsid surface [34–36]. This conformational change exposes the hydrophobic VP2 and VP3 internal capsid proteins that allow the virions to penetrate the ER membrane and reach the cytosol [37–39]. Once in the cytosol, they undergo disassembly to allow for transport through the narrow NPC and into the nucleus [10,40]. Importantly, previous studies have reported essential roles for VP3 in PyV ER-to-cytosol escape as well as viral disassembly in the cytosol [9,10]. Without a VP3, MCPyV may require different cellular factors to complete one or more of these upstream steps. During mitosis, the individual identities of the ER and nuclear envelope are lost, allowing ER components to segregate into daughter cells alongside nucleoplasmic proteins

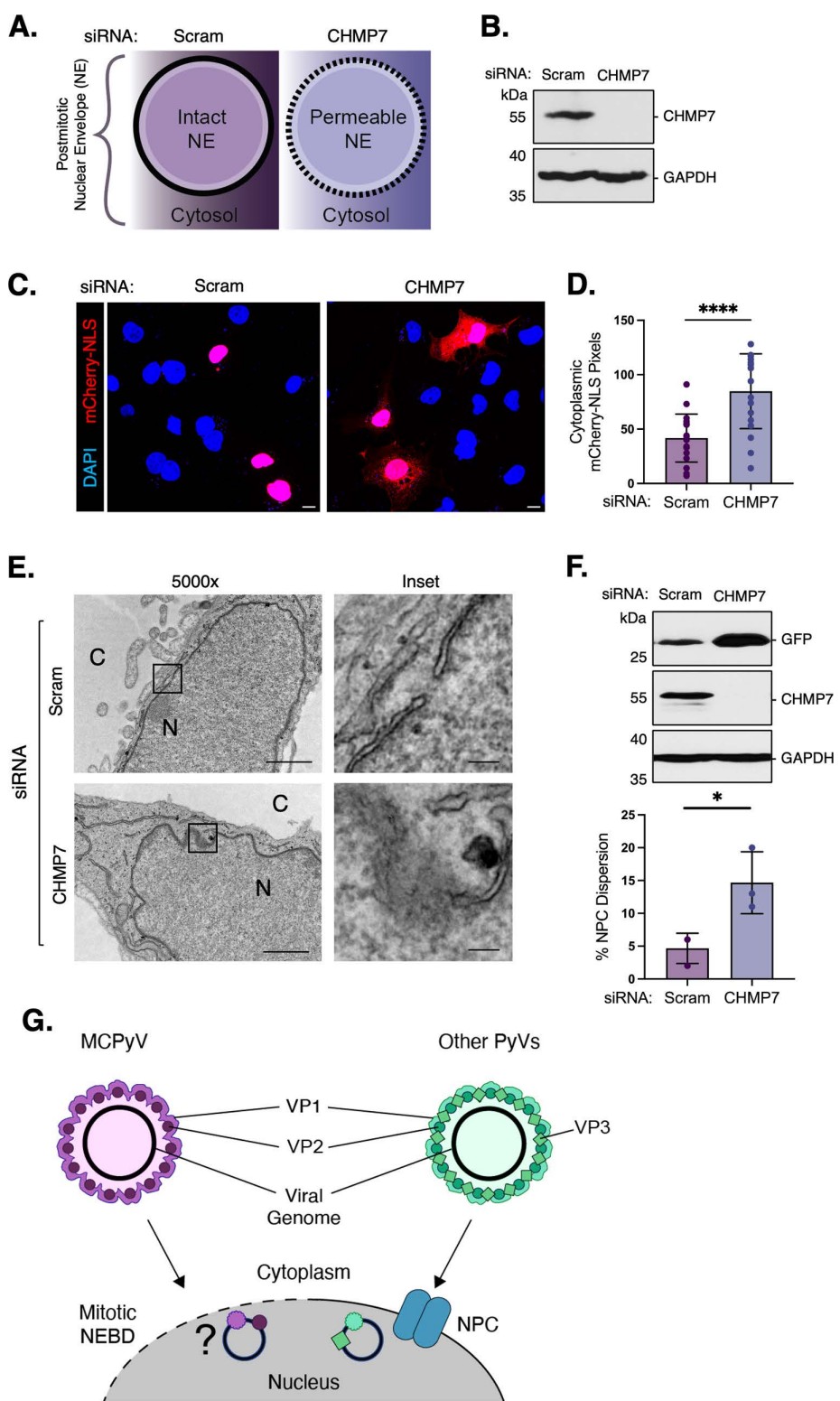

**Fig 5. Increasing membrane permeability is sufficient to support MCPyV nuclear entry.** (A) Schematic illustrating CHMP7's role in nuclear enve-
lope reformation. (B) COS-7 cells were transfected with 50 nM of either scrambled control siRNA (Scram) or siRNA against CHMP7. CHMP7 protein
levels were assessed by immunoblotting and GAPDH was used as a loading control. (C) Confocal microscopy of COS-7 cells that were transfected with

50 nM of either scrambled control siRNA (Scram) or siRNA against CHMP7 for 24 h and then mCherry-NLS for an additional 24 h. Cells were then fixed and counterstained with DAPI (blue). Scale bars: 10 μm. (D) Quantification of (C) was performed using EBImage in R with adaptive thresholding (n = 3, 50 cells/replicate). (E) TEM images of nuclear envelope morphology in COS-7 cells that were transfected with either scrambled control siRNA (Scram) or siRNA against CHMP7. Scale bars: 1000 nm (left), 100 nm (inset). (F) As in (B), except cells were infected with MCPyV-GFP for 48 h. GFP and CHMP7 protein levels were assessed by immunoblotting and GAPDH was used as a loading control. Values represent means ± SD from at least three independent experiments normalized to the loading control. Statistical significance was determined using an unpaired two-tailed Student's t-test (***p ≤ 0.001, **** p ≤ 0.0001).

[41]. Therefore, MCPyV might await mitotic NEBD to both escape the ER compartment and to enter the nucleus. If MCPyV can escape the ER in interphase cells, it may not engage with the downstream disassembly machinery, which would result in a viral particle that is too large to pass through the NPC. Immunofluorescence microscopy and cellular fractionation assays to determine where the virus is 'trapped' when the cell cycle is arrested will further clarify this question.

Intriguingly, our data show that VP1 alone is sufficient to traffic MCPyV into the nucleus during entry. A previous study reported similar findings demonstrating that MCPyV VP2 was dispensable for PsV infection in certain cell types [8]. This was not the case for BKPyV, where VP1 only PsVs exhibited a dramatic decline in infectivity [8]. They hypothesized that the larger MCPyV VP1 capsid (~61 aa) may have assumed some of the functions of the other PyV VP2/3 minor capsid proteins. Several other 'VP3-less' PyVs exist (all with larger VP1 proteins), although MCPyV is currently the only human PyV in this clade [8]. From our data, several other questions regarding the role of the MCPyV capsids in nuclear entry have emerged. First, is whether VP1 enters the nucleus along with the viral genome during infection. Previous work on SV40 showed that VP1 and VP3, but not VP2, accompany the viral genome into the nucleus [11] but the structural organization of nuclear-localized MCPyV is not yet known. Immunofluorescence experiments that track viral DNA for the presence or absence of MCPyV capsids inside of the nucleus may help to answer this question. Next, how does VP1 interact with host cellular machinery to traffic the virus to the nucleus? Similar to other DNA viruses, it likely uses the host cytoplasmic dynein motor protein-1 to traverse the intracellular space. Investigating the association between VP1 and components of the dynein motor complex is an important next step in developing a deeper understanding of the overall MCPyV entry pathway. Lastly, what is the role of VP2 in the viral life cycle and is it, in fact, cell type specific? Our data suggest that VP2 is not required for nuclear entry but perhaps it has a role in viral egress instead. As more human PyVs are discovered, structural and genetic studies characterizing the role of the MCPyV VP1 and VP2 capsid proteins in viral entry (and the overall life cycle) will become increasingly important.

Our findings add MCPyV to the list of cancer-causing viruses that exploit mitotic NEBD for nuclear entry, which also includes human papillomavirus (HPV) and murine leukemia virus [22,24]. With HPV, incoming viral particles are targeted to the nucleus through tethering of the L2 minor capsid protein to mitotic chromatin [23,42,43]. It remains unclear whether MCPyV uses a similar mechanism to localize to the nucleus during entry. SV40 is targeted to the nucleus through direct binding to Nesprin-2 on the outer nuclear membrane, where it is then handed off to the NPC for nuclear entry [11]. As Nesprin-2 also associates with mitotic chromatin during cell division [44], MCPyV may use a similar targeting mechanism as other PyVs to reach the nuclear membrane where it awaits NEBD instead of coordinating subsequent entry through the NPC. Future studies to investigate the molecular interactions between MCPyV capsid proteins and host cellular factors, such as Nesprin-2, could ultimately help to identify novel therapeutic targets against both the primary MCPyV infection and its associated malignancies.

## Materials and methods

### Cell culture

A549 (CCL-185), CV-1 (CCL-70) and COS-7 (CRL-1651) cells were obtained from ATCC. HEK 293TT cells were a gift from Dr. Christopher Buck (National Cancer Institute). Non-immortalized normal human dermal fibroblasts were a gift from Dr. Robert Kalejta (University of Wisconsin-Madison). All cells were cultured in Dulbecco's Modified Eagle's Medium (DMEM; ThermoFisher Scientific, 11965–118) supplemented with 10% (vol/vol) fetal bovine serum (FBS; R&D Systems, S11150). Cultures were maintained at 37 °C with 5% $CO_2$ and 95% humidity. All cell lines are mycoplasma negative.

## Generation of MCPyV pseudovirions

MCPyV pseudoviruses were prepared according to [13]. Briefly, HEK293TT cells were transfected with MCPyV VP1, VP2, and a reporter plasmid (pcDNA3.1) expressing GFP using polyethylenimine (PEI, Polysciences Inc.). After 72 h, cells were harvested and resuspended in lysis buffer (10% Triton X-100, 1 M $(NH_4)_2SO_4$, 0.01% Pen/Strep, 0.5 mM $MgCl_2$, 0.1 mM $CaCl_2$). Nuclease treatment was performed by adding 1 µL each of benzonase and exonuclease, followed by overnight incubation at 37°C to allow for non-encapsulated DNA degradation. Matured pseudovirion preparations were then loaded onto a discontinuous OptiPrep gradient (27%, 33%, and 39%) and centrifuged at 45,000 rpm for 4 hours at 4°C using an SW 50.1 rotor (Beckman Coulter). Purified virus was collected by gravity flow, and PsV concentration determined by SDS-PAGE and Coomassie staining compared to BSA standards. PsV preparations typically contain between 0.5-1µg VP1 protein/mL.

## Antibodies

MCPyV VP1 and VP2 antibodies were provided by Dr. Christopher Buck (Center for Cancer Research, National Cancer Institute), SV40 VP2/3 (Abcam ab53983), GFP (Takara 632381), GAPDH (Sigma CB1001-500G), Hsp90 (Santa Cruz sc-13119), IPO7 (Abcam ab99273), NUP98 (Cell Signaling 2598S), CHMP7 (Abcam ab242221), Nesprin-2 (Bethyl A305-393A).

## Plasmids

MCPyV VP1 (pwM) and VP2 (ph2m) were gifts from Dr. Christopher Buck (Addgene plasmid # 22515 and # 22518, respectively). MCPyV VP1-ΔNLS was generated by Vector Builder. BKPyV VP1, VP2, and VP3 were gifts from Dr. Michael J. Imperiale (University of Michigan). pmCherry-C1 mCherry-NLS was a gift from Dyche Mullins (Addgene plasmid # 58476). GFP pcDNA3.1(-) EGFP-FLAG was described previously [45].

## Pharmacological treatments and cell cycle synchronization

For pharmacological inhibition of importin-β, 10 µM of importazole (Millipore Sigma, SML0341) was added to cells 1 h prior to infection and kept on through the duration of the experiment. For cell cycle arrest, either 2 µM of aphidicolin (Millipore Sigma, A0781) or 9 µM Ro-3306 (Thermo Fisher, 418110) was added to cells 24 h before infection and kept on through the duration of the experiment. Cell cycle arrest was released by washing cells twice with fresh growth media and replacing it with media containing dimethyl sulfoxide (DMSO; Millipore Sigma, D2650).

## siRNA transfections

For all knockdown experiments, COS-7 cells were reverse transfected with 50 nM of the indicated siRNA using Lipofectamine RNAiMAX Transfection Reagent (ThermoFisher Scientific, 13778150) and Opti-MEM reduced serum media (Invitrogen, 31985). The following siRNAs were used in this study: siNUP98 (GUGAAGGGCUAAAUAGGAA) and siIPO7 (GCAAUAUAUGGCUCCUCGA) were synthesized by Sigma. siCHMP7 SMARTpool siRNA duplexes (CGACCUUGGU-AAACGGAAA, GGGUUUAUCCUGUCGCUAA, GGAGGUGUAUCGUCUGUAU, GUAACAAAUGGCUUAGAUU) were purchased from Dharmacon. All Star Negative (Qiagen, 1027281) was used as a scrambled control siRNA. Infections and biochemical assays were all performed at 24–48 h post transfection.

## Quantification of intracellular GFP

Cells were infected with MCPyV-GFP and harvested at the indicated timepoints. GFP reporter plasmid concentration and copy number were measured using quantitative PCR (qPCR). A standard curve was generated using serial dilutions (0–1 ng/µL) of the GFP pcDNA3.1 plasmid and qPCR reactions were performed using the SYBR Green Universal Master Mix (Roche) with the following primers: eGFP-F:

5'-AGTCCGCCCTGAGCAAAGA-3', eGFP-R: 5'-TCCAGCAGGACCATGTGATC-3', GAPDH-F: 5'-TCAAGGCTGAGAA CGGGAAG -3', and GAPDH-R: 5'-CGCCCCACTTGATTTTGGAG -3'. Primer efficiency was 88% based on dilution curve and efficiency calculations. Plasmid copy numbers were calculated using the formula: copies/ng = concentration (ng/μL) × $1.85 \times 10^{14}$, based on the molecular weight of the 5 kb plasmid ($3.25 \times 10^6$ Da). Sample analysis revealed copy numbers ranging from $2.80 \times 10^{13}$ to $5.10 \times 10^{20}$ across tested volumes (0–20 μL), with corresponding Cq values between 38.87 and 13.43.

## Transmission electron microscopy (TEM)

For TEM analysis, purified pseudoviruses were adsorbed onto formvar-coated copper grids (Electron Microscopy Sciences) for 45 sec, briefly rinsed with distilled water, and negatively stained with 2% uranyl acetate solution. Samples were then examined using a JEM-1400Plus electron microscope operating at an acceleration voltage of 80 kV and images were acquired using a TVIPS (Tietz Video and Image Processing Systems GmbH) 2K × 2K CMOS camera.

## Immunofluorescence microscopy

Cells were fixed with 1% paraformaldehyde for 15 min and then permeabilized with 0.2% Triton X-100 for 5 min at room temperature. Samples were blocked with 5% milk containing 0.02% Tween-20 and incubated with primary antibodies in blocking solution overnight at 4°C. Coverslips were washed 3x in milk and incubated with Alexa Fluor secondary antibodies (Invitrogen) for 30 min at room temperature. Coverslips were again washed and mounted on glass slides using ProLong Gold Antifade Mountant containing DAPI (Thermo Fisher, P36931). Imaging was taken using a Leica SP5 Lightning confocal microscope.

## Flow cytometry

Cells were washed twice with PBS and resuspended in a staining solution containing 10 μg/mL Hoechst 33342 (Thermo Fisher, 62249). Cells were incubated for 60 min at 37°C in the dark. Flow cytometry was performed using a ZE5 Cell Analyzer (Bio-Rad) equipped with a 405-nm laser for Hoechst excitation. A minimum of 10,000 events were collected per sample. Data were analyzed using FlowJo.

## Western blotting and quantification

Cells were lysed in RIPA buffer (50 mM Tris-HCl, pH 7.2, 150 mM NaCl, 0.1% Triton X-100, 1% sodium deoxycholate, and 5mM EDTA), incubated on ice for 30 min, and centrifuged at 21,3000 x g for 10 min at 4°C to remove the insoluble fraction. After boiling at 95°C for 10 min, 10–30 μL of each sample was separated using SDS-PAGE and transferred to Nitrocellulose Amersham Protran Premium Western blotting membrane (Millipore Sigma, GE10600003). Membranes were washed in tris-buffered saline (TBS) with 0.5% Tween-20 (TBST, EMD Millipore, 9480) 3 times for 5 min. Blots were then probed with primary antibodies at 4°C on a rocker overnight. Secondary antibodies from Jackson ImmunoResearch were diluted 1:5000 and HRP substrate (Millipore Sigma, WBULP) added for 3 min before membranes were exposed to HyBlot CL Audoradiography Film (Denville, E3018) and developed using a Konica Minolta SRX-101A Medical Film Processor. All western blots were quantified using FIJI. Signals were normalized to the whole-cell lysate GAPDH or Hsp90 signal. At least three independent replicates were quantified for each experiment, and statistical significance was determined using a standard Student's t test.

## Supporting information

**S1 Fig. MCPyV-ΔNLS VP1 fails to form pseudovirions.** (A) Plasmid map of the VP1 expression construct. (B) Amino acid sequences of VP1 (wild-type) and VP1-ΔNLS, showing the mutations that were introduced in the predicted nuclear localization signal (NLS). (C) Immunofluorescence microscopy of cells transfected with VP1 or VP1-ΔNLS constructs.

Scale bars: 10µm. (D) Transmission electron microscopy (TEM) images of purified MCPyV-GFP and MCPyV-ΔNLS preparations, showing normal pseudovirion formation with wild-type VP1 but not VP1-ΔNLS. Scale bars: 20 nm. (E) Coomassie-stained gel of purified virus preparations demonstrating the presence of VP1 protein in MCPyV-GFP but not MCPyV-ΔNLS samples. (F) COS-7 cells infected with equal amounts of MCPyV-GFP or MCPyV-ΔNLS pseodvirus from fractions 2–4 for 72 h. GFP expression indicates successful infection.
(TIFF)

**S2 Fig. Flow cytometry analysis of cell cycle synchronization.** Flow cytometry analysis of Hoechst-labeled cells confirming cell cycle arrest in experiments presented in Fig 4 of the main text. Histograms show DNA content distribution across G0/G1, S, and G2/M phases for each synchronization condition.
(TIFF)

## Acknowledgments

We thank members of the Spriggs and Tsai laboratories at the University of Michigan for thoughtful discussions throughout this work and the Imperiale laboratory at the University of Michigan for sharing BKPyV reagents. We also thank Dr. Christopher Buck at the National Cancer Institute's Laboratory of Cellular Oncology for providing MCPyV reagents.

## Author contributions

**Conceptualization:** Karen Wang, Adrienne N. Eady, Isabel Amaya, Chelsey C. Spriggs.

**Data curation:** Karen Wang, Adrienne N. Eady, Isabel Amaya, Alina Stanczak, Chelsey C. Spriggs.

**Formal analysis:** Karen Wang.

**Funding acquisition:** Chelsey C. Spriggs.

**Investigation:** Karen Wang, Adrienne N. Eady, Isabel Amaya, Chelsey C. Spriggs.

**Methodology:** Karen Wang, Adrienne N. Eady, Isabel Amaya, Chelsey C. Spriggs.

**Project administration:** Chelsey C. Spriggs.

**Supervision:** Chelsey C. Spriggs.

**Validation:** Karen Wang, Chelsey C. Spriggs.

**Visualization:** Karen Wang, Chelsey C. Spriggs.

**Writing – original draft:** Karen Wang, Chelsey C. Spriggs.

**Writing – review & editing:** Karen Wang, Adrienne N. Eady, Isabel Amaya, Chelsey C. Spriggs.

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
