## [Decision Letter · Decision Letter 0]

22 Jan 2025

Temporal gating of nuclear import: How Merkel cell polyomavirus exploits the cell cycle for nuclear entry

Dear Dr. Spriggs,

Thank you for submitting your manuscript to PLOS Pathogens. After careful consideration, we feel that it has merit but does not fully meet PLOS Pathogens's publication criteria as it currently stands. Therefore, we invite you to submit a revised version of the manuscript that addresses the points raised during the review process.

Please submit your revised manuscript within 60 days Mar 23 2025 11:59PM. If you will need more time than this to complete your revisions, please reply to this message or contact the journal office at plospathogens@plos.org. Please include the following items when submitting your revised manuscript:

We look forward to receiving your revised manuscript.

Kind regards,

Denise A Galloway

Academic Editor

PLOS Pathogens

Alison McBride

Section Editor

PLOS Pathogens

Editor-in-Chief

PLOS Pathogens

orcid.org/0000-0003-2946-9497

Michael Malim

Editor-in-Chief

PLOS Pathogens

orcid.org/0000-0002-7699-2064

**Additional Editor Comments:**

Your manuscript was well regarded by all three reviewers. They applauded that it was well written and used multiple approached to reach the conclusion that MCPyV, unlike other polyomaviruses uses nuclear membrane breakdown to enter the nucleus.. Among their comments are two that I feel need experiments to firm up the conclusions. The first is the choice of COS7 cell lines for most of the experiments. Since it expresses SV40 T antigen and is of monkey origin it would be best to use primary human dermal fibroblasts for at least one of the experiments with inhibitors or a knockdown. Secondly, there was a question about the requirement of VP1 for entry. please use the GFP plasmid on its own and pseudo virus that has been treated with DNase. These experiments will make an even stronger case for the proposed mechanism of entry.

**Journal Requirements:**

2) When completing the data availability statement of the submission form, you indicated that you will make your data available on acceptance. We strongly recommend all authors decide on a data sharing plan before acceptance, as the process can be lengthy and hold up publication timelines. Please note that, though access restrictions are acceptable now, your entire data will need to be made freely accessible if your manuscript is accepted for publication. This policy applies to all data except where public deposition would breach compliance with the protocol approved by your research ethics board. If you are unable to adhere to our open data policy, please kindly revise your statement to explain your reasoning and we will seek the editor's input on an exemption. Please be assured that, once you have provided your new statement, the assessment of your exemption will not hold up the peer review process.

3) Please ensure that the funders and grant numbers match between the Financial Disclosure field and the Funding Information tab in your submission form. Note that the funders must be provided in the same order in both places as well. State what role the funders took in the study. If the funders had no role in your study, please state: "The funders had no role in study design, data collection and analysis, decision to publish, or preparation of the manuscript.".

If you did not receive any funding for this study, please simply state: u201cThe authors received no specific funding for this work.u201

**Reviewers' Comments:**

Reviewer's Responses to Questions

**Part I - Summary**

Reviewer #1: This is beautifully written manuscript and is an important contribution to the field. The Sprigg's group presents compelling data proving that MCPyV, unlike other polyomaviruses, delivers the viral genome (or what suffices for a genome) to the nucleus following nuclear membrane breakdown. Genetic and biochemical approaches were used and the experiments were very well-controlled. MCPyV is unique from the other polyomaviruses in that it lacks VP3 and also lacks a nuclear localization signal in VP2, both of which are important for nuclear penetration. The question addressed in how does this virus then deliver its genome to the nucleus. They tested several possibilities using using a pseudovirus system that packages a GFP expressing plasmid upon entry into the nucleus. This is somewhat of a surrogate system for studying infection so the word transduction should replace infection throughout the manuscript, in my opinion. The experiments involved inhibiting nuclear pore mediated import with drugs and specific genetic knockdowns. These treatments had no effect on MCPyV but did inhibit transduction by BK pseudoviruses. Chimeric pseudoviruses containing MCPyV VP1 and BK VP2 and 3 were used and this virus was also resistant to inhibitors of nuclear pore mediated import. A pseudovirus containing only MCPyV VP1 was sufficient and they conclude that VP1 is driving the process of nuclear entry. Is it possible, however, that the genome escapes the capsid before nuclear entry and it is the genome that enters and not VP1 + genome? A genome alone control might answer this question. Manipulation of the cell cycle and another knockdown that partially permeabilizes the nuclear membrane made it clear that this virus unlike other polyomaviruses does not penetrate through an intact nuclear membrane but rather waits for mitotic breakdown of the nuclear membrane, similar to what happens during papillomavirus infection. The only control missing was a GFP plasmid alone control. Is it possible that the pseudoivirus preparations contain both packaged GFP plasmid and unpackaged plasmids that are able to enter the cell and penetrate. Treating the preparations with DNase would eliminate this possibility?

In summary, this an elegant and well-controlled study that sheds light on the cell biology of MCPyV infection of cells.

Reviewer #2: In the research article by Wang et al., the authors investigate the nuclear entry process of Merkel cell polyomavirus, a small DNA tumor virus that causes Merkel cell carcinoma. They use MCPyV pseudoviruses, composed of the late viral structural proteins VP1 and VP2 and a GFP reporter construct in place of the viral genome, to infect COS-7 cells and subsequently study viral entry dynamics with a focus on nuclear entry steps. The expression of GFP is therefore used a surrogate marker for nuclear entry of MCPyV. Using a combination of pharmacological inhibitors and siRNA knockdown approaches, the authors report an unconventional method by which MCPyV enters the nucleus. Contrary to other well-studied polyomaviruses such as SV40 and BKPyV, MCPyV does not appear to require the nuclear pore complex as a major entry mechanism. Instead, the authors present data to support the use of cell cycle-dependent nuclear envelope breakdown to deliver genetic material into nucleus. They provide evidence that this process is facilitated by the viral VP1 protein, which again is unique to MCPyV as the nuclear entry process for other polyomaviruses usually involve the VP2/VP3 proteins.

This study focuses on an area of MCPyV virology that is currently poorly characterized. It is of interest to those in the MCPyV and DNA tumor virus fields, as well as to the broader field of those studying host-pathogen interactions involved in viral entry dynamics. Overall, it is well-organized and well-written. The discussion is thought-provoking and well-considered. The methodology is appropriate for their research objectives and the experiments include relevant controls. However, there are several areas where the methodology and approaches could have been expanded to strengthen/elevate their findings and better support their mechanistic interpretations in order to meet the high standards for publication in PLOS Pathogens.

Reviewer #3: The manuscript by Wang et al., entitled “Temporal gating of nuclear import: How Merkel cell polyomavirus exploits the cell cycle for nuclear entry” reveals how MCPyV enters the nucleus by taking advantage of nuclear envelope breakdown during mitosis. The authors present an elegant analysis showing that entry of the MCPyV pseudovirions require either a leaky nuclear envelop or break down during mitosis for the MCPyV pseudovirus to enter the nucleus based on transcription and translation of a reporter gene. BKPyV pseudoviruses were able to enter the nucleus through the nuclear pores and this served as a negative control. It is nice that they showed EM of the pseudoviruses. The study is well done and is sigificant because it shows that MCPyV nuclear entry is different from other polyomaviruses. This work will be broadly interesting to all virologists. This reviewer has only identified some minor issues with this clearly presented manuscript.

**Part II – Major Issues: Key Experiments Required for Acceptance**

Reviewer #1: None

Reviewer #2: Major areas for improvement to support study conclusions are outlined below.

1. The choice of cell lines to test and the ultimate use of only one cell line throughout this study lessens the impact of these findings. Given that the tropism of the human MCPyV virus is likely cutaneous, it is intriguing that the authors chose a lung epithelial cell line and monkey kidney fibroblasts for these studies instead of skin-derived keratinocytes and fibroblasts. The use of at least two human cell lines (whether epithelial or fibroblasts) would make these findings more compelling and relevant to the natural tropism of the virus.

2. The use of multiple approaches (pharmacological inhibition with IPZ, siRNA to IPO7, NUP98, and CHMP7, electron microscopy, etc) to interrogate nuclear entry is a strength of this manuscript. The authors interpret their findings that IPZ treatment and NUP98 knockdown significantly reduced infection to mean that a subset of viral particles may still use the NPC for nuclear entry. Given the questions surrounding the role of and necessity for the MCPyV VP1 NLS in nuclear entry, studies using NLS mutagenesis would likely provide opportunities to further clarify the mechanistic role, if any, of the MCPyV VP1 NLS in nuclear entry. Perhaps there is a reason for not doing so (i.e. structural fidelity of capsid). Regardless, the mutagenesis of VP1 NLS would strengthen their findings in addition to drug and KD studies.

3. In Figure 5C, there appears to be a mixed population of cells in which a subset has increased membrane permeability following successful CHMP7 knockdown. The use of a GFP reporter allows visual inspection of cells that achieved successful nuclear entry. Could the authors use co-IF to determine if NEBD in cells with CHMP7 knockdown (and cytoplasmic mCherry) is, indeed, coincident with GFP expression on a per-cell basis? Or did they only quantify intracellular GFP in the whole population (Figure 5F)? Such analysis would provide strong evidence (on a per-cell basis) to support their hypothesis that increased nuclear membrane permeability facilitates infection.

Reviewer #3: None.

**Part III – Minor Issues: Editorial and Data Presentation Modifications**

Reviewer #1: A GFP plasmid alone control may shed light on when genome is released from the capsid. Does VP1 enter along with genome or is the genome released first?

I suggest using transduction rather than infection given that these are pseudoviruses.

Also, why are COS7 cells more susceptible to this? Could it have to do with the fact that they express SV40 T antigen that causes the cells to cycle more rapidly?

Reviewer #2: Other areas for improvement and consideration:

1. The investigators initially tested three different cell lines for infection with MCPyV PsV (A549, CV-1, and COS-7). Of the three lines tested, COS-7 supported the most robust infection and nuclear entry. Given the current ambiguity surrounding MCPyV tropism, the authors could have leveraged these differences in infectivity to explore whether cellular tropism is in any way determined by the nuclear entry mechanism they discovered contributes to infectivity. For instance, does CHMP7 knockdown in A549 and CV-1 cells now make them permissive for infection/GFP expression? Is there a qualitative or quantitative difference between epithelial cells (A549) and fibroblasts (CV-1)? Such studies could potentially expand and elevate the impact of these studies.

2. The header “COS-7 cells support infection” should be more specific. The other cell lines also support ‘infection,’ just at a lower level than COS-7. Also, it seems the authors can only say that cells support PsV infection as this is not a natural MCPyV infection system.

3. Figure 1C: What are the concentrations of pseudoviruses used for infection? The Materials and Methods section indicates that the PsV stock concentration is determined, but the figure only indicates they are using volume (10-20uL) to infect. Was the percentage of cells that are infected calculated (using GFP immunofluorescence for instance)?

4. In Figure 4, was activity of their inhibitors of cell cycle progression (aphidocolin and RO-3306) confirmed to affect the cell cycle as predicted in each experiment? Or was it only performed in the independent experiment shown in Figure 4B? Given the extent to which they are interpreting these results, it would be useful to include these controls in every experiment, either by flow cytometry or some other biomarkers of cell cycle stage, to validate the inhibitors are functioning as expected.

5. Materials & Methods

a. siRNA sequences for CHMP7 and IPO7 need to be included

b. do specific catalog numbers and application-specific dilutions for antibodies need to be listed?

c. need to add methods sections for Western blotting (how were lysates made, membrane, protein concentration loaded, transfer buffer, etc) and PsV infection (are cells prepared for infection, is a specific media used, is viral inoculum replaced or left on cells, etc).

Reviewer #3: Dharmacon should provide SMARTpool siRNA sequences for publication if you reach out to them and these should be included in the methods.

There is some concern about quantifications of the westerns from film as the methods are not clearly stated whether HRP is used, which may not provide for a quantitative analysis of protein levels over what appears to be a broad range. Secondary antibodies were not mentioned.

The authors also need to report the efficiency of the real-time qPCR reactions.

In figure legend 1E (lines 466-467) the authors need to clarify if they are quantifying the GFP reporter mRNA or DNA plasmid.

The authors need to make it clear in the figure legends whether pretreatment with drug extends past and during viral infection, see figure 2.

In some of the figures the labels are too small and would be easier to read if they were larger fonts. Also, the fonts between the westerns and the graphs are not the same. Examples are figure 4 graphs and experimental diagrams, 2, 5D, 5F, 3D, 3F and 1E. Probably removing the bold font and increasing the font size would help a lot.

PLOS authors have the option to publish the peer review history of their article (what does this mean? ). If published, this will include your full peer review and any attached files.

**Do you want your identity to be public for this peer review?** For information about this choice, including consent withdrawal, please see our Privacy Policy .

Reviewer #1: No

Reviewer #2: No

Reviewer #3: No

**Figure resubmission:**

**Reproducibility:**



---

## [Editor Report · Decision Letter 1]

15 May 2025

Dear Dr. Spriggs,

We are pleased to inform you that your manuscript 'Temporal gating of nuclear import: How Merkel cell polyomavirus exploits the cell cycle for nuclear entry' has been provisionally accepted for publication in PLOS Pathogens.

Best regards,

Denise A Galloway, PhD

Academic Editor

PLOS Pathogens

Alison McBride

Section Editor

PLOS Pathogens

Sumita Bhaduri-McIntosh

Editor-in-Chief

PLOS Pathogens

orcid.org/0000-0003-2946-9497

Michael Malim

Editor-in-Chief

PLOS Pathogens

orcid.org/0000-0002-7699-2064

The authors have successfully addressed the criticisms.
---

## [Editor Report · Acceptance letter]

Dear Dr. Spriggs,

We are delighted to inform you that your manuscript, "Temporal gating of nuclear import: How Merkel cell polyomavirus exploits the cell cycle for nuclear entry," has been formally accepted for publication in PLOS Pathogens.

Best regards,

Sumita Bhaduri-McIntosh

Editor-in-Chief

PLOS Pathogens

orcid.org/0000-0003-2946-9497

Michael Malim

Editor-in-Chief

PLOS Pathogens

orcid.org/0000-0002-7699-2064